# The Periodic Response of Tidal Flat Sediments to Runoff Variation of Upstream Main River: A Case Study in the Liaohe Estuary Wetland, China

**Haifu Li [1,2,3,†], Lifeng Li [3,4,†], Fangli Su [2,3,*], Tieliang Wang [2,3] and Peng Gao [1,*]**

1    College of Forest, Shandong Agricultural University/Mountain Tai Forest Ecosystem Research Station of State Forestry and Grassland Administration, Tai An 271018, China; syaulhf@syau.edu.cn
2    College of Water Conservancy, Shenyang Agricultural University, Shenyang 110866, China; tieliangwang@126.com
3    Liaoning Shuangtai Estuary Wetland Ecosystem Research Station of State Forestry and Grassland Administration, Panjin 124112, China; lihaifulinghai@126.com
4    College of Science, Shenyang Agricultural University, Shenyang 110866, China
*    Correspondence: fanglisu8@163.com (F.S.); gaopengy@163.com (P.G.); Tel.: +86-139-4026-0797 (F.S.); +86-138-5484-7572 (P.G.)
†    Contributed equally to this work.

**Abstract:** (1) Background: To reveal the intrinsic relationship between the tidal flat sediments in an estuary wetland and the runoff from the upstream river. This research was conducted in the tidal flats of the Liaohe estuary wetland. (2) Methods: The $^{137}$Cs and $^{210}$Pb dating technique was used to reconstruct the time correspondence between tidal flat sediments and runoff, and the periodic response was explored between the changes in the tidal flat sediments and runoff based on the spectrum analysis method. (3) Results: The average sedimentation rate in the tidal flat was 2.24 cm·year$^{-1}$ during the past 50 years. The amount of fine sediment particles deposited on the estuary tidal flat was directly related to the amount of sediments transported by the river and inversely proportional to the ability of rivers to transport fine matter. The high frequency reproduction cycle of 14–15 years in the flood season flow and 5–6 years in the annual sediment discharge of the Liaohe River correspond to the high and low frequency reproduction cycles of the median size of sediments in Liaohe estuarine wetland tidal flats. (4) Conclusions: The research clarified the hydrological constraints of the action law between Liaohe River runoff and the estuarine sediments. The periodic response between Liaohe River runoff and the sediment was established.

**Keywords:** tidal flat; estuary; sediment; runoff and sediment discharge; periodic response

## 1. Introduction

The ocean is the final destination of fine-grained matter from land, and estuarine wetlands are an important land–sea transition zone. Frontal tidal flats of estuarine wetlands are the interface of the land–sea exchange of materials, where sediment deposition is the most sensitive [1]. The sediment characters of estuary tidal flats are affected by the changes in runoff and sediment transportation directly from upstream rivers. Regarding the relationship between sediment deposition in estuaries and the hydrological processes of rivers, it is very important to establish the time correspondence between the tidal flat sediment deposition parameters and the time series data of rivers, such as runoff and sediment discharge. Therefore, establishing this correspondence requires the identification of the environmental information carried by the fine sediments deposited in the tidal flats of the estuary [2].

A variety of measures have been used in sedimentary history research, including horizon markers, anchored tiles, rulers, sediment traps, optical backscatter sensors, short-lived radionuclides, biomarkers, and magnetic minerals [3–5]. The basis for the ability to record changes in sediment deposition time in all the methods depends on the availability of time markers in the sediment column [6]. Sediment fingerprinting is a well-established method that has proven its value in revealing the time dimension in sediment research [7,8]. Among the methods mentioned above, the molecular diffusion of $^{137}$Cs in sediment may change the vertical profile of the sediment, but it is impossible to change the position of its accumulation peak [9,10]. Therefore, the application of $^{137}$Cs provides an effective way to quantitatively study sedimentary history. $^{137}$Cs is an artificial nuclide with two major peak fallouts in 1963 and 1986 as time markers in the northern hemisphere [11–13]. The $^{210}$Pb dating technique can also be used to document changes in sedimentation rates though time and provide a basis for establishing changes in sedimentation process over the past ca. 100 years [14]. In the actual dating analysis, the $^{210}$Pb$_{ex}$ (half-life 22.3a) specific activity in the sediment has been used to estimate the sedimentary age and deposition rate [15,16]. The $^{137}$Cs and $^{210}$Pb dating techniques have been successfully used in related areas for the historical information of sediments in China [17–20]. In addition, the use of radioactive isotopes $^{137}$Cs and $^{210}$Pb has provided historical sedimentary information for tidal flat sediments in estuaries, which has solved the problem of consistent time scales for exploring the relationship between downstream sediments and the runoff of upstream rivers [21].

River sediment discharge is the main material source for tidal flats in estuary wetlands extending to the sea [22]. The tidal flat sedimentation state changes due to changes in runoff and sediment discharge from rives [23]. These changes can be determined by analyzing changes in the sediment characteristics [24]. Studies have confirmed that the relative proportions of different particulate matter in sediments are closely related to runoff properties [25]. Some studies have suggested that the amount of fine-grained sediments is directly related to the amount of sediments transported to the river and inversely proportional to the power of the river to transport fine-grained sediments [26]. The sediment carried by runoff has a preferential flow of finer particles during migration and sedimentation. Coarser particles are preferentially deposited during transport, and the distribution is uniform along the direction of runoff transport [27,28]. The sediment deposited in the estuary is mostly composed of loamy sand or sandy loam. During the flood period, coarse-grained sediment can be transported to the shoal sediments in the estuary due to the high runoff energy. As a result, the sediment particle size becomes coarser [29]. Research in Fukushima, Japan showed that the changes in the sediment grain size in the upper sediments of a seasonal submerged sandbank occurred because the sediment component transported by the runoff was changed [30]. The changes in river runoff processes have a direct impact on sedimentation processes in tidal flats in downstream estuaries. There is an inevitable connection between river runoff and estuarine sediments. However, the hydrological constraints on the relationship between river runoff and estuarine sedimentation are still unclear. Studies on the periodic relationship between river runoff and sediment have rarely been reported.

The Liaohe estuary wetland is an internationally important wetland. It is located at the core of the Bohai Bay and is an important channel for the Liaohe River sediments to enter the sea. The frontal tidal flat of the Liaohe estuary wetland is an important material exchange zone in the land–sea interlaced area of Liaodong Bay, China. Sediment deposition is very complicated in this zone and is affected by natural and man-made factors [31]. The sedimentation process and the interaction between tidal flat sediments and upstream rivers are still unclear in this area. There is an urgent need to clarify the intrinsic relationship between sediment deposition in the tidal flat of the Liaohe estuary wetland and the runoff of the Liaohe River. Therefore, the objectives of this study are to: (1) clarify the sedimentary characteristics of the tidal flat; (2) explore the relationship between the tidal flat sediments and river runoff; and (3) determine the periodic relationship between river runoff and sediments.

## 2. Materials and Methods

### 2.1. Study Area

This study was conducted in the tidal flats at the boundary between the land and the sea of the Liaohe estuarine wetland, located in Panjin City, Liaoning Province, in Northern China (Figure 1). This area is an important material exchange zone in Liaodong Bay, and the sedimentation process is affected by runoff from the Liaohe River, ocean tides and human factors. The main material source for the continuous tidal flat deposition in the Liaohe estuary wetland is the sediment discharge of the Liaohe River. The study area is in a warm temperate continental semi-humid monsoon climate zone with four distinct seasons, with rain and heat in the same season, dry and cold in the same period, an annual average temperature of 8.40 °C, an annual average precipitation of 623.20 mm, and an annual average evaporation of 1669.60 mm. According to the data of the Resource and Environment Data Cloud Platform of the Chinese Academy of Sciences (http://www.resdc.cn/data.aspx?DATAID=283), the main sediment source contribution area of the Liaohe estuary wetland tidal flat is the main stream basin of Liaohe River. During 1985 to 2017, the area of cultivated land, sandy land, construction land, and beaches and tidal flats increased by 4.36%, 84.0%, 85.0% and 29.5%, respectively, while the forest land, grassland, and water area decreased by 4.96%, 46.72%, and 9.52%, respectively [32].

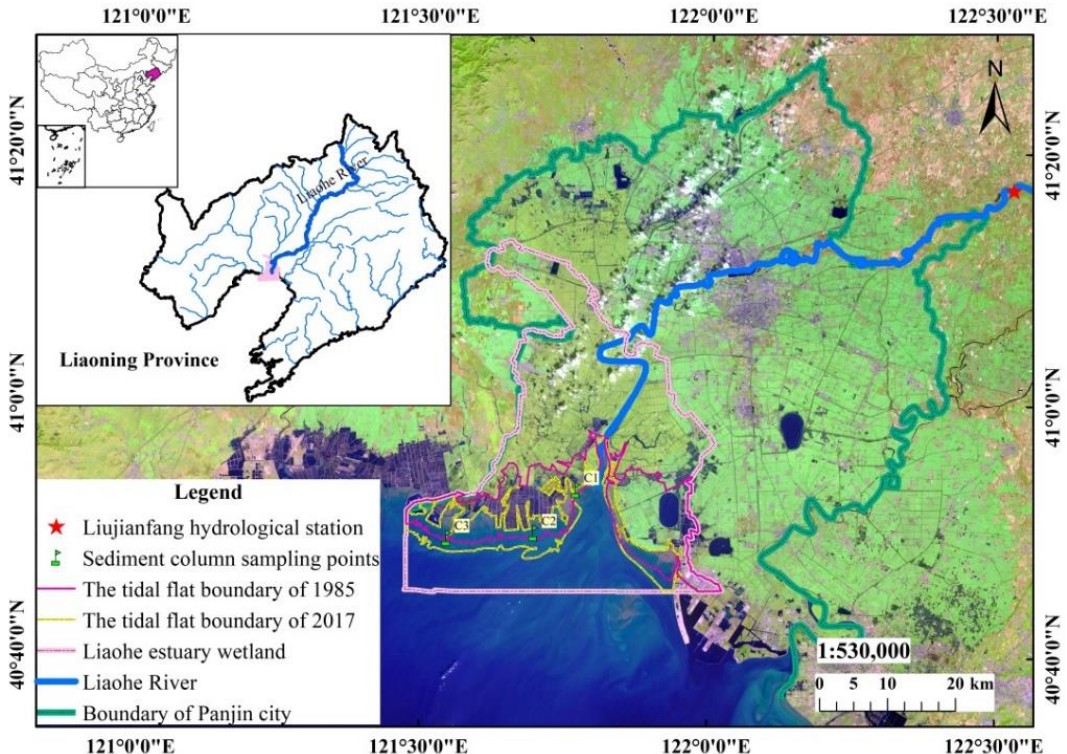

**Figure 1.** The location of the study area.

### 2.2. Sample Collection

To obtain information on both the sedimentary history and sediment deposition rate of the tidal flats in the Liaohe estuarine wetland, three vertical sediment columns (C1, C2, C3) were sampled randomly in the central area of the tidal flats using a gravity sediment sampler on 16 May 2017 (Figure 1). The sampling points were located in the core area of the Liaohe Estuary National Nature Reserve, China. Human activity is not allowed across the entire region, therefore the tidal flat was formed naturally and there has been no human disturbance in the past 50 years or more.

According to the deposition rate of the northern part of Liaodong Bay and Gaizhou Beach, over nearly 30 years [33,34] the sampling depth of the sediment column was determined to be

140 cm. Due to the high moisture content of sediments, it was not appropriate to split the samples on site. Therefore, the sample columns were sent to the laboratory for freezing immediately after they were collected. The samples were cut at regular intervals after 24 h of storage. The surface layer was subdivided at a resolution of 1.5 cm from 0–60 cm and a resolution of 2 cm from 61–140 cm, which referenced the sample segmentation methods from related studies [20,25]. The divided samples were air-dried according to the corresponding number of layers and the air-drying period was 90 days. In the last 30 days, the sample weights were measured every 10 days, and the constant weight of each sample was regarded as the weight that did not change, and the weight was accurate to 0.01 g. After air-drying, the samples were sieved using a 1.50 mm sieve and then sealed for [137]Cs and [210]Pb activity testing.

The testing of the sediment particle size used the same column that was used to test [137]Cs and [210]Pb. The preparation process was the same as the preparation of the [137]Cs and [210]Pb sample testing. The samples were sieved by using a 1.50 mm sieve after air-drying and then dispersed for particle size testing. The particle sizes were measured with a Matersizer 3000 laser particle sizer (Malvern, UK) with a particle size range of 0.20–3000.00 µm. The sediment components and particle size parameters were statistically analyzed and categorized using the Folk–Ward method [35].

### 2.3. Data Analysis

### 2.3.1. Sedimentary Dating

The sedimentary history was determined by the [137]Cs and [210]Pb dating techniques. The [137]Cs and [210]Pb activities were performed using a high purity germanium detector gamma spectrometer (BE5030, HPGe, San Ramon, CA, USA) with a system energy resolution of 2.25 KeV. The test time of each sample was 249,456 s to 56,342 s to ensure that the measurement error was controlled below 6%.

(1)　[137]Cs dating technique

The year was calibrated by analyzing the [137]Cs-specific activity peak. The sediment's deposition time information was determined based on the [137]Cs nuclear accumulation precipitation peak in the Northern Hemisphere, which appeared in 1963, and the Chernobyl nuclear leakage secondary accumulation peak, which appeared in 1986 [36,37].

The deposition rate was calculated using the following formula:

$$D_r = \frac{H}{T_1 - T_2} \tag{1}$$

In the formula, $D_r$ is the deposition rate (cm·year$^{-1}$), $T_1$ and $T_2$ are the years specified by the half-life dating method (year); and $H$ is the time slice depth or depth difference between two time horizons (cm).

(2)　[210]Pb dating technique

The specific activity of [210]Pb$_{ex}$ at the bottom of the sediment sample did not reach the base value. In addition, the study area was an estuary tidal flat, where the sediment mainly came from upstream erosion sediment transportation. Therefore, the CIC model was used to conduct a dating analysis based on [210]Pb$_{ex}$. The constant initial concentration model of [210]Pb (CIC) assumed that the sediment 210Pb$_{ex}$ has a constant initial concentration in the model. The initial specific activity of 210Pb$_{ex}$ defined at the sediment-water interface was $A_0$, and the specific activity of [210]Pb$_{ex}$ at a certain sediment mass depth m was $A_m$. The $A_m$ has an exponential decay relationship with the increase of mass depth [38]. The formulas were as follows:

$$A_m = A_0 e^{-\lambda t} \tag{2}$$

$$t = \frac{1}{\lambda} \ln \frac{A_0}{A_m} \tag{3}$$

$$S = \frac{m}{t} \tag{4}$$

In the formulas, $A_0$ is the initial specific activity of $^{210}$Pb$_{ex}$ at the sediment–water interface; $A_m$ is the specific activity of $^{210}$Pb$_{ex}$ at mass depth $m$ (Bq·g$^{-1}$); $\lambda$ is the decay coefficient of $^{210}$Pb, 0.03114·year$^{-1}$; $m$ is the mass depth which refers to the cumulative value of sediment above a certain depth (g·cm$^{-2}$); $t$ is deposition time (y); $s$ is the mass deposition rate in time $t$ (g·(cm$^2$·year)$^{-1}$). For the verification of the deposition rate determined by the $^{137}$Cs, the mass deposition rate (g (cm$^2$·year)$^{-1}$) was converted to the deposition rate (cm·y$^{-1}$) according to the sediment bulk density and porosity correction.

2.3.2. Relationship between the Sediment Deposition Characteristics and Runoff

Based on the dating results, the age-related information of the tidal flat sediments in the Liaohe estuary wetland was determined through sedimentation rate inversion, and the time–correspondence relationship between the estuary tidal flat sediments and the Liaohe River runoff was reconstructed. The relationship between the tidal flat sediments and runoff was explored using the time–correspondence relationship. The mean size, median size, clay content and $^{137}$Cs specific activity were selected to characterize the sedimentary features of the estuarine tidal flat sediments. The specific surface area of the sediments was selected as an auxiliary index. The main index of the flow in the flood season, the mean annual flow, and the annual sediment discharge, which are close to the sediment transportation of the Liaohe River, were selected as the key indicators. The above runoff indicator data were obtained from the "Liaohe River Hydrological Yearbook" during the period from 1987–2015. Data from the Liujianfang Hydrological Station were cited in this study, which is the last hydrological station before the Liaohe River inflow to the estuary [39], located in Taian County, Anshan City, China, 63 km from the Liaohe estuary wetland.

The key hydrological indicators were measured and calculated as follows:

(1) The flow in the flood season: This refers to the river flow in the relatively concentrated rainfall season of each year, which was calculated according to the daily flow data of the river hydrological observation section. The flood season of Liaohe River Basin ranged from June to September in each year. The formula was as follows:

$$F_f = \frac{\sum\limits_{i=1}^{n} F_i}{n} \tag{5}$$

where $F_f$ is the flow in the flood season (m$^3$·s$^{-1}$), $F_i$ is daily average flow rate (m$^3$·s$^{-1}$), and $n$ is the days of the flood season (D).

(2) The mean annual flow rate: This refers to the average value of river flow during each year, which was calculated based on the daily flow data of river hydrological observation sections. The formula was as follow:

$$F_m = \frac{\sum\limits_{i=1}^{N} F_i}{N} \tag{6}$$

where $F_m$ is the mean annual flow rate (m$^3$·s$^{-1}$), $F_i$ is daily average flow rate (m$^3$·s$^{-1}$), and $N$ is the days of each year (D).

(3) The annual sediment discharge: This refers to the total sediment transported through the river hydrological observation section in one year, which was calculated from the daily sediment concentration data of river hydrological observation sections. The formula was as follows:

$$S = \sum_{i=1}^{N} A_i \times Q_i \tag{7}$$

where $S$ is the annual sediment discharge (t·year$^{-1}$), $A_i$ is daily average sediment concentration (kg·m$^{-3}$), $Q_i$ is daily runoff (m$^3$), and $N$ is the number of days in a year (D).

All the daily data were measured by automatic observation device, and then the mean daily data were automatically generated.

The correlation matrix of the Pearson correlation coefficient was used to analyze the correlations between hydrological indicators of Liaohe River and sediment particle size index. The analysis was completed in Matlab2010b software.

### 2.3.3. Periodic Relationship between Sediment and Runoff

The cycle reproducibility on the flow in the flood season, the mean annual flow, the annual sediment discharge of the Liaohe River runoff index and the mean and median sizes of the deposited sediments in the Liaohe estuary tidal flats were determined by spectrum analysis, which used a fast Fourier transform to explore the periodicity of the data. According to the spectral analysis results, the frequency corresponding to the peak values in the maximum power, the maximum secondary power and the maximum third power were selected to obtain the periodicity of each index.

The main calculation process is as follows:

Step 1.　Centralize the original data and perform a fast Fourier transform (FFT) to turn the original data $x_t$ into $X_T(f)$.

Step 2.　Calculate the power spectral density.

$$P(f) = \frac{\left| X_T(f) \right|^2}{T} \tag{8}$$

According to $X_T(f) = A + JB$ ($A$ is the real part, $B$ is the imaginary part), the power spectral density calculation formula can be simplified as:

$$P(f_i) = \frac{A_i{}^2 + B_i{}^2}{T} \tag{9}$$

where $f_i$ is the line frequency and T is the length of the data interval.

The line frequency $f_i = i/T = i/N, i = 0, 1, 2, \cdots, N-1$; $N$ is the amount of the data.

Step 3.　Drawing the spectrogram

The spectrogram will be plotted based on the calculated power spectral density $P(f_i)$ and line frequency $f_i$. The period will be calculated according to the frequency corresponding to the peak value of the spectrum, and the period is the reciprocal of the frequency.

## 3. Results

### 3.1. Sedimentary History of the Tidal Flats in the Liaohe Estuary Wetland

(1)　$^{137}$Cs dating technique

According to the linear quasi-determination of sediment bulk density and sedimentation depth of sediment column C1, C2 and C3, the sediment bulk density shows a linear increasing trend with the increase of sediment depth ($R^2$ $_{C1}$ = 0.8391, $R^2$ $_{C2}$ = 0.8543, $R^2$ $_{C3}$ = 0.8812,). So the $^{137}$Cs profile peak value was suitable for the dating analysis. The average $^{137}$Cs-specific activity of the sediment sample column was 1.10 Bq·kg$^{-1}$ (Figure 2(C1)). There were two complete accumulation peaks with increasing depth increases: the maximum peak was at 120.00 cm with a $^{137}$Cs-specific activity of 2.47 Bq·kg$^{-1}$, and the sub-maximum peak was at 78.00 cm with a $^{137}$Cs-specific activity of 1.52 Bq·kg$^{-1}$. Consequently, it was determined that the $^{137}$Cs accumulation peak at 120 cm was 1963, and the $^{137}$Cs accumulation peak at 78 cm was 1986. The sediment deposition rate was 1.83 cm·y$^{-1}$ from 1963 to 1986, the deposition rate was 2.52 cm·year$^{-1}$ from 1986 to 2017, and the average deposition rate was 2.22 cm·year$^{-1}$ from 1963 to 2017.

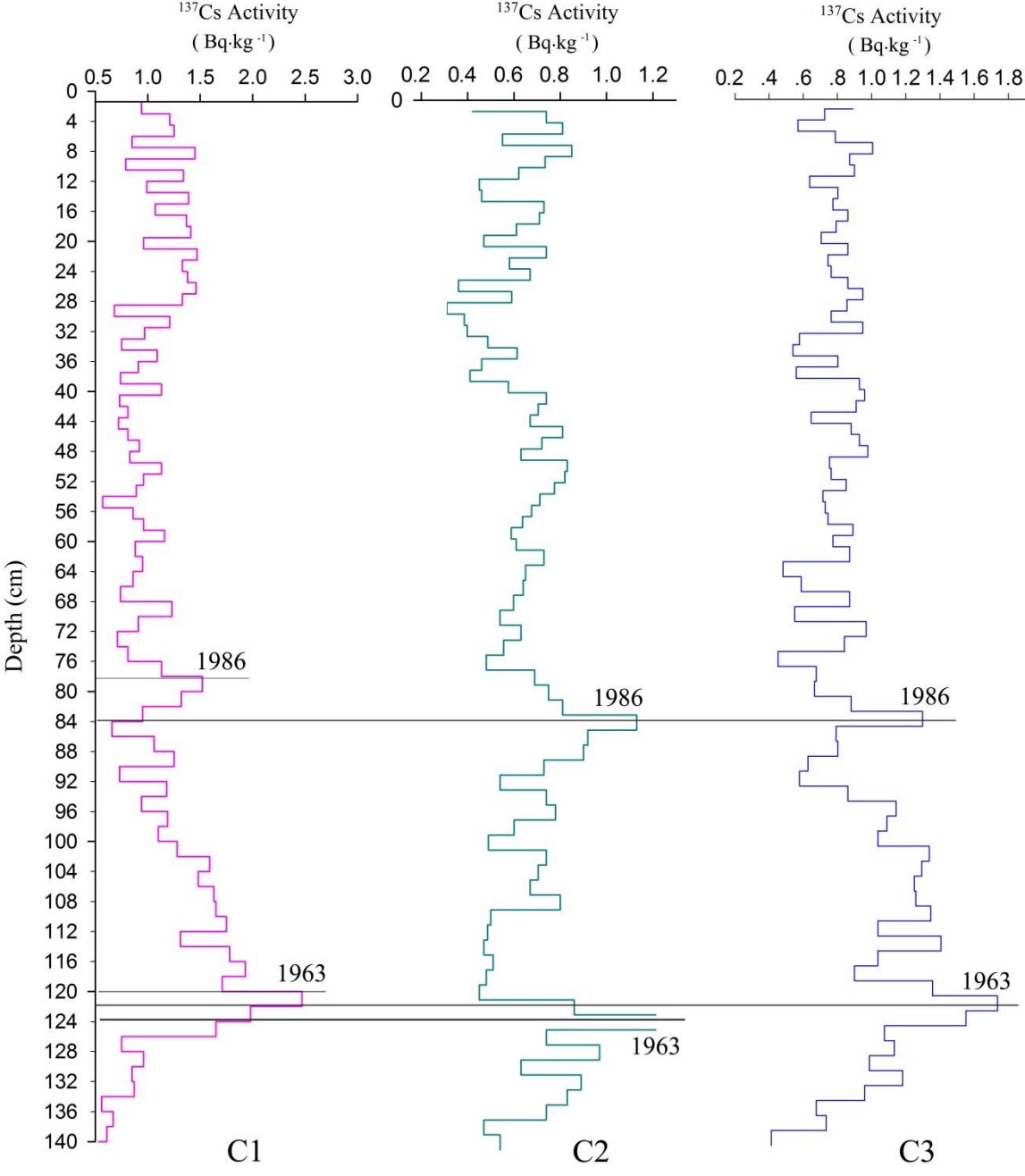

**Figure 2.** The dating analysis results of $^{137}$Cs.

Likewise, we can see from (Figure 2(C2)) that the maximum peak of $^{137}$Cs-specific activity was at 124.00 cm and the sub-maximum peak was at 84.00 cm. The sediment deposition rate was 1.74 cm·year$^{-1}$ from 1963 to 1986, the deposition rate was 2.71 cm·year$^{-1}$ from 1986 to 2017, and the average deposition rate was 2.29 cm·year$^{-1}$ from 1963 to 2017. Similarly known, the maximum peak of $^{137}$Cs-specific activity was at 124.00 cm and the sub-maximum peak was at 84.00 cm based on the data analysis of Figure 2(C3). The sediment deposition rate was 1.65 cm·year$^{-1}$ from 1963 to 1986, the deposition rate was 2.71 cm·year$^{-1}$ from 1986 to 2017, and the average deposition rate was 2.26 cm·y$^{-1}$ from 1963 to 2017.

(2) $^{210}$Pb dating technique

According to the linear quasi-determination of the ln($^{210}$Pb$_{ex}$) value and sedimentation depth of sediment column C1, C2 and C3, the ln($^{210}$Pb$_{ex}$) shows a linear decreasing trend with the increase of

sediment depth ($R^2$ $_{C1}$ = 0.8985, $R^2$ $_{C2}$ = 0.7871, $R^2$ $_{C3}$ = 0.8048,). So, the CIC model was suitable for the $^{210}Pb_{ex}$ dating analysis. The deposition time of the entire sediment sample column C1 from the surface layer to the bottom layer was 2017 to 1953, which has a 64-year sediment record (Figure 3(C1)). The sediment deposition rate was 2.21cm·y$^{-1}$ from 1963 to 1986, the deposition rate was 2.50 cm·y$^{-1}$ from 1986 to 2017, and the average deposition rate was 2.22 cm·year$^{-1}$ from 1963 to 2017 based on the $^{210}Pb$ dating result.

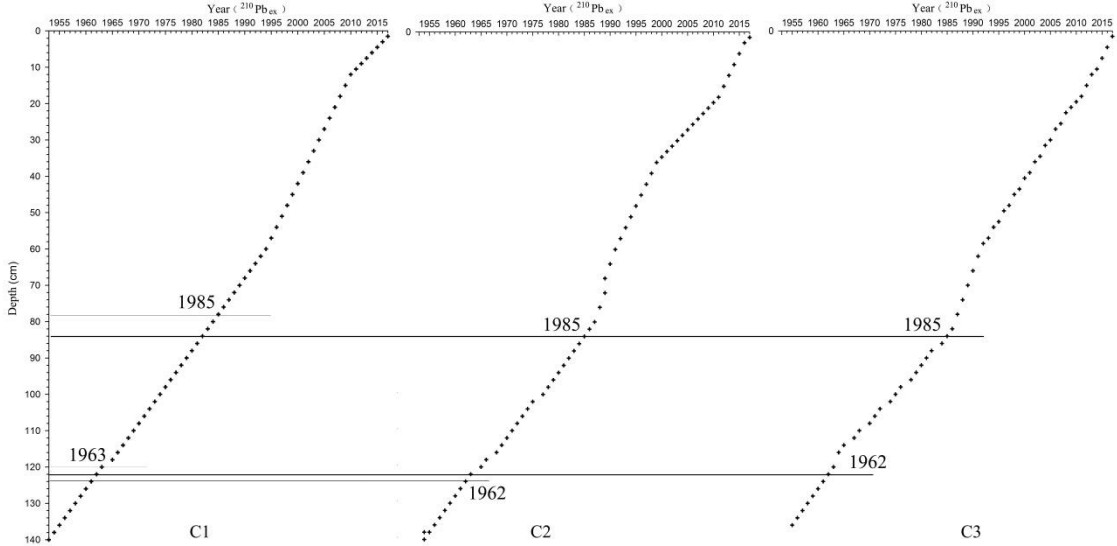

**Figure 3.** The dating analysis results of $^{210}Pb_{ex.}$

Likewise, the sediment deposition rate was 1.74cm·y$^{-1}$ from 1963 to 1986, the deposition rate was 2.65 cm·y$^{-1}$ from 1986 to 2017, and the average deposition rate was 2.26 cm·y$^{-1}$ from 1963 to 2017 based on $^{210}Pb$ dating analysis of Figure 3(C2). The sediment deposition rate was 1.65cm·y$^{-1}$ from 1963 to 1986, the deposition rate was 2.65cm·y$^{-1}$ from 1986 to 2017, and the average deposition rate was 2.22 cm·y$^{-1}$ from 1963 to 2017 based on $^{210}Pb$ dating analysis of Figure 3(C3).

Combining the $^{137}Cs$ and $^{210}Pb_{ex}$ dating results, the chronological time determined by the two methods mutually corresponded. The dating results of sediment sample columns C1, C2 and C3 are as follows. C1: The dating results at the depth of 120 cm determined by the two methods were identical, both in 1963. The dating result of $^{137}Cs$ at the depth of 78 cm was 1986, and the dating result of $^{210}Pb_{ex}$ was 1985. C2: The dating results of $^{137}Cs$ at the depth of 124 cm and 84 cm were 1963 and 1986, respectively, and the dating results of $^{210}Pb_{ex}$ were 1962 and 1985. C3: The dating results of $^{137}Cs$ at the depth of 122 cm and 84 cm were 1963 and 1986, and the dating results of $^{210}Pb_{ex}$ were 1962 and 1985. According to the above analysis results, the dating results of the two methods have a one-year relative deviation which showed a slight deviation. Overall, the dating analysis results were accurate and reliable.

### 3.2. Relationship between Sediment Deposition in the Tidal Flats of the Liaohe Estuary Wetland and the Runoff from the Liaohe River

There is a consistent correlation between the mean size, median size, clay content, specific surface area and $^{137}Cs$-specific activity of the sediment granularity parameters and the flow in the flood season, the mean annual flow, and the annual sediment discharge from the Liaohe runoff index (Figure 4).

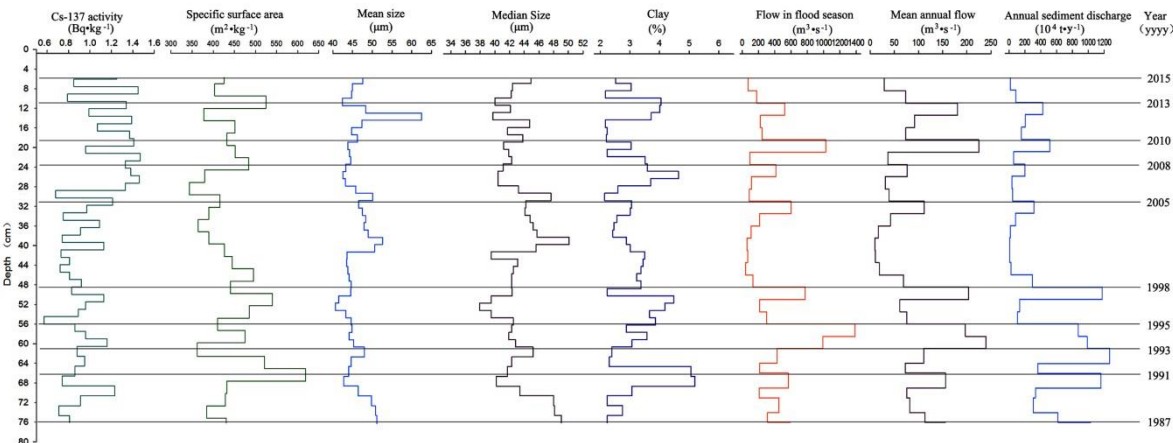

**Figure 4.** Relationship between the sediment grain size characteristics and the runoff index based on dating analysis.

According to the measured hydrological data from Liujianfang Hydrological Station on the Liaohe River, 2010, 1998, 1995, 1993, 1991 and 1987 had higher amounts of runoff and sediment transportation. The minimum values of the flow in the flood season, the mean annual flow and the annual sediment discharge were more than 500.00 m$^3$·s$^{-1}$, 180.00 m$^3$·s$^{-1}$ and 5.00 million t·year$^{-1}$ respectively among these years. The corresponding depth of the deposited sediment in these years had mean sizes greater than 43.00 μm, with an average value of 46.07 μm and a maximum value of 51.13 μm. The corresponding median sizes were greater than 41.00 μm, with an average value of 43.63 μm and a maximum value of 49.03 μm. The corresponding clay contents were less than 3.00%, except in 1991, and the average clay content was 2.97%. The corresponding $^{137}$Cs-specific activities were less than 1.00 Bq·kg$^{-1}$, except in 2010, and the average value was 0.85 Bq·kg$^{-1}$.

The years 2013, 2008 and 2005 had a medium amount of runoff and sediment discharge. The values of the flow in flood season, the mean annual flow and the annual sediment discharge were in the range of 400.00 to 600.00 m$^3$·s$^{-1}$, 100.00 to 180.00 m$^3$·s$^{-1}$ and 2.00 to 4.30 million t·y$^{-1}$ respectively among these years. The mean sizes of the corresponding sediment depths in these years were between 42 μm and 46.00 μm, and the average value was 44.03 μm. The corresponding median sizes were between 40.00 μm and 44.00 μm, and the average value was 41.53 μm. The average mean size and median size decreased by 2.04 μm and 2.10 μm, respectively, compared to the average values in the higher runoff and sediment discharge years. The corresponding clay contents were between 3.00% and 4.00%, and the average value was 3.61%, which increased 0.63% compared to the average value in the higher runoff and sediment discharge years. The corresponding $^{137}$Cs-specific activities showed increasing trends in 2013 and 2005, except in 2008, with an average value of 1.29 Bq·kg$^{-1}$, which increased 0.32 Bq·kg$^{-1}$ compared to the average value in the higher runoff and sediment discharge years.

The other years had low flows and sediment discharge. The maximum values of the flow in the flood season, the mean annual flow and the annual sediment discharge were less than 400.00 m$^3$·s$^{-1}$, 100.00 m$^3$·s$^{-1}$ and 2.00 million t·y$^{-1}$ respectively among those years. The average mean sizes of the corresponding sediment depths in those years was 46.69 μm, and the average median size was 43.83 μm, which increased 0.43 μm and 0.19 μm compared to the average values in the higher runoff and sediment discharge years and increased 2.46 μm and 2.29 μm compared to the average values in the medium runoff and sediment discharge years. The corresponding average clay content was 2.80%, which decreased 0.18% compared to the average value in the higher runoff and sediment discharge years and decreased 0.81% compared to the average value in the medium runoff and sediment discharge years. The corresponding average $^{137}$Cs-specific activity was 1.02 Bq·kg$^{-1}$, which increased 0.05 Bq·kg$^{-1}$, which is close to the average value in the higher runoff and sediment discharge years, and decreased 0.11 Bq·kg$^{-1}$ compared to the average value in the medium runoff and sediment discharge years.

### 3.3. Sediment Deposition and Runoff Cycles

Based on the previous analysis of the corresponding relationship between the tidal flat sediments and the upstream river runoff and sediment discharge, the cyclical relationship between them will be deeply explored. There is clear cycle reproducibility in all the indexes of flow in the flood season, the mean annual flow and annual sediment discharge of the Liaohe River and the mean particle size and median size of the sediments in the Liaohe estuary wetland tidal flats (Figure 5 and Table 1). On a 30 year time scale, there was a 14 to 15 year recurrence cycle, a 3 to 4 year recurrence cycle and a 2 to 3 year recurrence cycle of the flow in the flood season. The cycle reproducibility of the mean annual flow and the annual sediment discharge were consistent, with a recurrence cycle of 5 to 6 years, a recurrence cycle of 3 to 4 years and a recurrence cycle of 2 to 3 years. Within a sedimentary depth range of 30 years, there was a recurrence cycle of 14 to 15 years, a recurrence cycle of 7 to 8 years and a recurrence cycle of 4 to 5 years for the mean particle size. There was a recurrence cycle of 14 to 15 years, a recurrence cycle of 5 to 6 years, and a recurrence cycle of 3 to 4 years for the median size. Overall, there was a correspondence cycle of 14 to 15 years between the mean particle size and the median size of the tidal flat sediments in the Liaohe estuary wetland and the flow in the flood season in the Liaohe River. There were 5 to 6 year and 3 to 4 year correspondence cycles between the median size of the tidal flat sediments, the mean annual flow and the annual sediment discharge of the Liaohe River. The cycle of 4 to 5 years in the mean particle size of the tidal flat sediments was close to the cycle of 5 to 6 years in the mean annual flow and the annual sediment discharge of the Liaohe River.

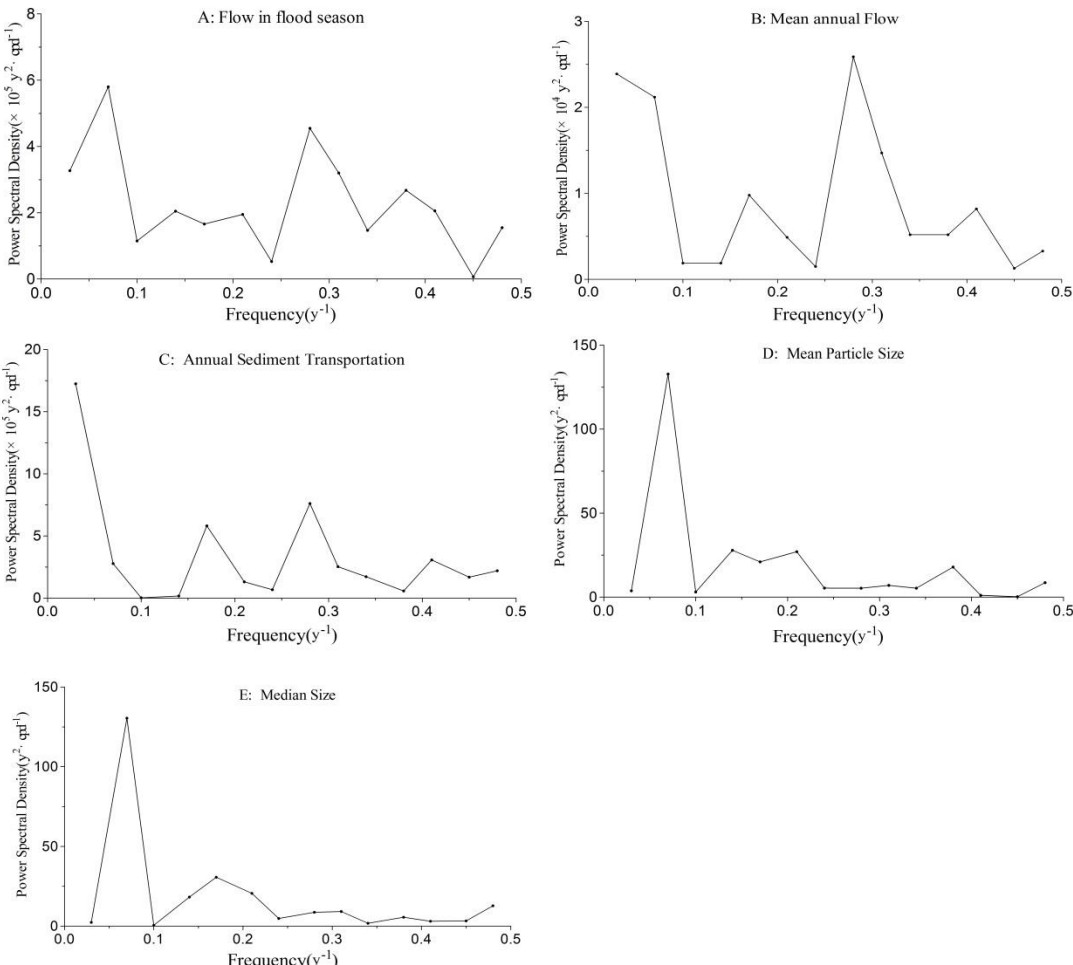

**Figure 5.** Spectrum analysis of the Liaohe River runoff index and the sediment particle size parameters of the tidal flat sediments in the Liaohe estuary wetland.

**Table 1.** Cycle analysis of Liaohe runoff index and sediment particle size parameters of tidal flat sediments in Liaohe estuary wetland.

| Indexes of Runoff Discharge | Frequency (year$^{-1}$) | Periodicity (year) | Indexes of Sediment Particle Size | Frequency (year$^{-1}$) | Periodicity (year) |
|---|---|---|---|---|---|
| Flow in flood season | 0.07 | 14.3 | Mean particle size | 0.07 | 14.30 |
| | 0.28 | 3.60 | | 0.14 | 7.10 |
| | 0.38 | 2.60 | | 0.22 | 4.50 |
| Mean annual flow | 0.17 | 5.90 | Median size | 0.07 | 14.30 |
| | 0.28 | 3.60 | | 0.17 | 5.90 |
| | 0.41 | 2.40 | | 0.31 | 3.20 |
| Annual sediment discharge | 0.17 | 5.90 | | | |
| | 0.28 | 3.60 | | | |
| | 0.41 | 2.40 | | | |

Meanwhile, according to the analysis of the spectrum power (Figure 5), the cycle corresponding to the high-power spectrum peak value ($5.8 \times 10^5$ a$^2$·cpd$^{-1}$) of the flow in the flood season was consistent with the cycle corresponding to the high-power spectral peak values (132.75 a$^2$·cpd$^{-1}$ and 130.41 a$^2$·cpd$^{-1}$) of the mean particle size and median size, which was a 14 to 15 year recurrence period. The cycles corresponding to the high-power spectrum peak values ($2.59 \times 10^4$ a$^2$·cpd$^{-1}$ and $7.62 \times 10^5$ a$^2$·cpd$^{-1}$) of the mean annual flow and the annual sediment discharge were consistent with the cycles corresponding to the high-power spectral peak values (30.69 a$^2$·cpd$^{-1}$) of median size, which was a 5 to 6 year recurrence period.

## 4. Discussion

In the present study, the deposition rate of the three sampling points was different, and this may be due to the fact that the dating results of [137]Cs and [210]Pb were not absolute dates. Besides, the corresponding sedimentary time at the same mass depth was not the same, and resulted in different calculation results [40,41]. Furthermore, there was a one year difference between the dating results of [137]Cs and [210]Pb, which implied that the dating results of the two methods were basically consistent and credible. Additionally, the average deposition rate of the three sampling points ranged from 2.22 to 2.27 cm·y$^{-1}$, with the maximum difference of 0.05 cm·y$^{-1}$ and an average output value of 0.03 cm·y$^{-1}$, which indicated that there was little difference in the deposition rates, and the values were almost the same in the area. Thus, it was reasonable and accurate to use [137]Cs and [210]Pb for dating analysis and sediment deposition rate calculation. Therefore, the average deposition rates of the Liaohe estuary tidal flat determined by two methods were as follows: 2.24 cm·y$^{-1}$ during the period of 1963 to 2017 and 1.80 cm·y$^{-1}$ during the period of 1986 to 1963. Furthermore, the sedimentation rate was 2.62 cm·y$^{-1}$ in the Liaohe estuary wetland tidal flats from 1986 to 2017, which was close to results from the Gaizhou Beach area of the Liaohe estuary and the Liaohe estuary region of Northern Liaodong Bay [33,34]. The results indicated that the deposition rate in the tidal flats in the Liaohe estuary wetland identified in this study was consistent with the deposition rate in the larger regional environment of the Liaohe Delta. In addition, the deposition rate C2 was greater than C1 and C3, which may be caused by extreme rainfall storm events in the upstream basin [42–44]. For example, there were extreme flood events of the Liaohe River in 1998 and 1993. The greater runoff capacity could transport a large amount of sediment to the relatively far distance of sampling point C2, and resulted in little deposition in C1, while the larger distance of C3 lead to relatively little sediment. Therefore, there existed differences in the spatial distribution of the deposition rate within the area.

The sediment discharge from the Liaohe River was negatively correlated with the clay content of the tidal flat sediments, and positively correlated with the mean particle size and median size of the sediment, and significantly negatively correlated with the clay content and [137]Cs-specific activity of the sediments on a nearly 30-year time scale (Table 2). According to the results of related research, it

was believed that the amount of fine sediment particles deposited on the estuary tidal flat was directly related to the amount of sediments transported by the river and inversely proportional to the ability of rivers to transport fine matter [26]. In the Evrotas River research, in Greece, the sediment transportation to the downstream estuary was significantly reduced due to the decrease in runoff from the upstream river during the dry season [45]. It was shown that the sediments deposited in the Yangtze River estuary, China are significantly reduced in the context of the decline of the upper reaches of the river runoff and sediment discharge [46]. The studies have shown that the changes of river runoff directly affected the amount of sediment transported to the estuary, which was consistent with the findings of this study. Other research has shown that the proportion of coarse-grained sand in sediment carried by river runoff increased, and the ratio of clay content decreased as the flow of the upstream river increased, resulting in the increase of sediment particle size in an estuary in Fujian, Southeastern China [24]. In the Yellow River estuary research, in China, the estuary sediment particle size has an obvious coarsening trend due to the increase in upstream runoff and sediment transport by the implementation of the water and sediment adjustment project in the Yellow River [47,48]. In general, the fine-grained clay content tended to decrease, and the sediment particle size showed a coarsening trend of the estuary wetland tidal flats with the increase in sediment discharge of rivers, which were consistent with the findings of this study.

**Table 2.** Correlation analysis between hydrological indicators of Liaohe River and sediment particle size index.

| Index | $^{137}$Cs Activity | Specific Surface Area | Median Size | Mean Particle Size | Clay |
|---|---|---|---|---|---|
| Clay | 0.510 | 0.902 ** | −0.607 | −0.622 | 1 |
| Flow in flood season | −0.652 | −0.658 | −0.645 | −0.647 | −0.627 |
| Mean annual flow | −0.590 | 0.540 | −0.781 | −0.718 | −0.523 |
| Annual sediment discharge | −0.883 ** | −0.065 | 0.856 * | 0.847 * | −0.768 * |

* indicates significance at the 0.05 level, ** indicates significance at the 0.01 level.

There are few studies discussing the cyclical relationship between tidal flat sediments in estuary wetlands, river runoff and sediment transport. Some studies have shown that the sedimentary volume in the Yangtze River estuary, in China, and the runoff in the basin have a 9 to 11 year cycle correspondence [49]. The mean particle size of the sediments in the Pearl River estuary had a 2 to 4 year cycle correspondence with the Pearl River sediment transport in China [50]. However, due to notable differences between the hydrological characteristics of the Yangtze River, the Pearl River and the Liaohe River, the relevant results have little significance for the study of sediment deposition in the tidal flats of the Liaohe estuary wetland. Other studies have shown that river runoff has a significant cyclical effect on the tidal constant of the estuary. There are 2.50, 5.50 and 6.50 years of common movement and interdecadal cycle relationships in the La Plata estuary in Spain [51]. The study in Spain has not yet determined whether the river runoff and sediment transportation have a cyclical effect on the fine-sediment deposition in the estuary. The results in this research indicated that the long recurrence times of the flow in the flood season in the Liaohe River affected the high frequency recurrence of the grain size parameters of the estuary tidal flats. The long recurrence times of the mean annual flow and the annual sediment discharge affected the low frequency reproduction of the sediment particles. There is a 14 to 15 year cycle correspondence between the mean particle size and the median size of the tidal flat sediments in the Liaohe estuary wetland and the flow in the flood season in the Liaohe River. There is a 5 to 6 year cyclical and a 3 to 4 year cyclical correspondence between the median size of the tidal flat sediments and the mean annual flow and the annual sediment discharge of the Liaohe River.

## 5. Conclusions

In the present study, the dating results of $^{210}Pb_{ex}$ were consistent with that of $^{137}Cs$, which indicated that the determined sediment dating results were credible. The average sedimentation rate in the center of the Liaohe estuary wetland tidal flats was 2.24 cm·y$^{-1}$ over the past 50 years. The sedimentation rates from 1963 to 2017 and 1963 to 1986 were 2.62 cm·y$^{-1}$ and 1.80 cm·y$^{-1}$, respectively. Besides, this study clarified that the amount of fine-sediment particles deposited on the estuary tidal flat was directly related to the amount of sediments transported by the river and inversely proportional to the ability of rivers to transport fine matter. After the runoff sediment transport capacity was reduced to a certain extent, the flow in flood season, the mean annual flow and the annual sediment discharge ranged from 400.00 to 600.00 m$^3$·s$^{-1}$, 100.00 to 180.00 m$^3$·s$^{-1}$ and 2.00 to 4.30 million t·y$^{-1}$, respectively. Furthermore, soil particle size and median size decreased with the decreasing of annual sediment discharge, while the clay content of sediment fine particles increased with decreasing of annual sediment discharge. The sediment carrying capacity was limited when the flow in the flood season was less than 400.00 m$^3$·s$^{-1}$ and the mean annual flow was less than 100.00 m$^3$·s$^{-1}$. The total amount of fine sediments transported to the estuary tidal flats was reduced due to the lack of sediment transportation power of the runoff. In addition, there was a 14 to 15 year corresponding periodic relationship between the mean particle size and the median size of the tidal flat sediments and the flow in the flood season in the Liaohe River. There was a 5 to 6 year and a 3 to 4 year corresponding periodic relationship between the median size of the tidal flat sediments and the mean annual flow and the annual sediment discharge of the Liaohe River. The results will provide theoretical support for revealing the sedimentation and erosion balance of estuarine sediments and the stable maintenance of tidal flats.

**Author Contributions:** Writing—original draft preparation, H.L.; writing—review and editing, H.L., F.S.; investigation, H.L.; data curation, L.L., T.W.; supervision, P.G.; project administration, F.S., P.G. All authors have read and agreed to the published version of the manuscript.

**Funding:** This research was funded by the National Natural Science Foundation of China (31570706, 31670711, 31470710) the National Key Research and Development Program of China (2016YFC0500408, 2017YFC1503105); the Natural Science Foundation of Shandong Province of China (ZR2016CM49); and the Special Fund for Forestry Scientific Research in the Public Interest (201404303–08).

**Acknowledgments:** Thanks to Peng Gao, Fangli Su and Lifeng Li for their guidance in the experimental research and paper writing stages. Thanks to the members of the research team, Linlin Dong, Fei Song, Jian Cheng, Zichen Niu and Zhenhua Ding, for their assistance in the experimental process. The authors would like to thank the editor and referees for their invaluable suggestions.

**Conflicts of Interest:** The authors declare no conflict of interest.

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
