# Peer review of "The Periodic Response of Tidal Flat Sediments to Runoff Variation of Upstream Main River: A Case Study in the Liaohe Estuary Wetland, China"

_water, doi:10.3390/w12010061_

Round 1
Reviewer 1 Report
Paper is generally well written and I don't have any objections, except for CONCLUSIONS - please expand this chapter and support it a bit more with data. Now this chapter is very brief and not very well supported by data.Therefore I propose a minor revision.
Author Response
Responses to Reviewers’ Comments
Ref. No.: water-659177
Title: The periodic response of tidal flat sediments to runoff variation of upstream main River: A case study in the Liaohe estuary wetland, China
Journal name: water
Dear Reviewers,
Thank you very much for your careful review and constructive comments related to our manuscript. All suggestions and comments are helpful to improve the quality of our manuscript. We have studied all comments carefully and made a great effort to revise the manuscript accordingly. Once again we appreciate you and the reviewers’ efforts and time. If you have any question about this paper, please don’t hesitate to inform us as soon as possible.
We hope that the revised manuscript is suitable for publication.
Revised sections were marked in red color in the revised manuscript. The point-by-point responses to the editor’s and reviewers’ comments/questions are detailed below.
Yours sincerely,
Peng Gao and Haifu Li
15 December 2019
Responses to Reviewers’ Comments
1 Paper is generally well written and I don't have any objections, except for CONCLUSIONS - please expand this chapter and support it a bit more with data. Now this chapter is very brief and not very well supported by data. Therefore I propose a minor revision.
Responses: Thanks so much for pointing this out. We revised it accordingly in the section of conclusions. The revised descriptions are:
In the present study, the dating results of 210Pbex were consistent with that of 137Cs, which indicated that the determined sediment dating results were credible. The average sedimentation rate in the centre of the Liaohe estuary wetland tidal flats was 2.24 cm·y-1 in the past 50 years. The sedimentation rate from 1963 to 2017 and 1963 to 1986 were 2.62 cm·y-1 and 1.80 cm·y-1, respectively. Besides, this study clarified that the amount of sediment fine particles deposited on the estuary tidal flat was directly related to the amount of sediments transported by the river and inversely proportional to the ability of rivers to transport fine matter. After the runoff sediment transport capacity was reduced to a certain extent, the flow in flood season, the mean annual flow and the annual sediment discharge ranged from 400.00 to 600.00 m3·s-1, 100.00 to 180.00 m3·s-1 and 2.00 to 4.30 million t·y-1, respectively. Furthermore, soil particle size and median size decreased with the decreasing of annual sediment discharge, while the clay content of sediment fine particles increased with decreasing of annual sediment discharge. The sediment carrying capacity was limited when the flow in the flood season was less than 400.00 m3·s-1 and the mean annual flow was less than 100.00 m3·s-1. The total amount of fine sediments transported to the estuary tidal flats was reduced due to the lack of sediment transportation power of the runoff. In addition, there was a 14- to 15-year corresponding periodic relationship between the mean particle size and the median size of the tidal flat sediments and the flow in the flood season in the Liaohe River. There were a 5- to 6-year and a 3- to 4-year corresponding periodic relationship between the median size of the tidal flat sediments and the mean annual flow and the annual sediment discharge of the Liaohe River. The results will provide theoretical support for revealing the sedimentation and erosion balance of estuarine sediments and the stable maintenance of tidal flats.
Please see lines 375-395.
Thanks again for your good suggestions. We deeply appreciate your hard work for our manuscript and your comments and suggestions are valuable to improve the quality of our manuscript.

Reviewer 2 Report
・General Comments
The authors explore the relationship between sediment deposition rate and runoff characteristics of upstream river in the Liaohe estuary using radioactive isotope. The data are potentially interesting and worthy of eventual publication. On the other hand, explanation of the calculation method of correlation or hydrological variables was missing. While the findings presented in this study should be of interest to the audience, it is my opinion that a rather substantial revision, based on the technical comments given below, is needed to make this manuscript suitable for publication.
・Specific Comments
Abstract (Line 22)The unit of cm・a-1 is not generally used. Explanation of the abbreviation of “a” should be added.
Keywords (Line 30)
The word “cycle” is fairly abstract. The keyword should be more objective.
Materials and Methods
As the authors mentioned in line 96-97, sedimentation process in estuary is affected by human impact of upstream watershed and ocean process (wave and tide). Why authors deal with only upstream hydrology to evaluate the sedimentation process? The change of upstream area (land use or construction of cross drainage work) in the Liaohe river should be mentioned.
Materials and Methods (Line 108)
Is it right to recognize that the sampling points are not affected by any human influences? Condition of the sampling points should be described.
Materials and Methods (Line 126)
The reference of Folk-Ward method should be added.
Materials and Methods (Line 162-175)
Calculation method of hydrological indicators should be accurately described. For example, the period of flood season or measuring method of the sediment discharge.
Results (Line 246-252)
These sentences may move to discussion section.
Results (Line 330)
The calculation method of correlation showed in Table 2 should be described in 2. Materials and Methods section.
Results (Figure 3)
As the results of dating analysis of 210Pb, the deposition rate was different among the sampling points especially after 1985 (such as deposition rate of C2 1985-1990 was steeper than C1). The differences of deposition characteristic among stations should be discussed.
Author Response
Responses to Reviewers’ Comments
Ref. No.: water-659177
Title: The periodic response of tidal flat sediments to runoff variation of upstream main River: A case study in the Liaohe estuary wetland, China
Journal name: water
Dear Reviewers,
Thank you very much for your careful review and constructive comments related to our manuscript. All suggestions and comments are helpful to improve the quality of our manuscript. We have studied all comments carefully and made a great effort to revise the manuscript accordingly. Once again we appreciate you and the reviewers’ efforts and time. If you have any question about this paper, please don’t hesitate to inform us as soon as possible.
We hope that the revised manuscript is suitable for publication.
Revised sections were marked in red color in the revised manuscript. The point-by-point responses to the editor’s and reviewers’ comments/questions are detailed below.
Yours sincerely,
Peng Gao and Haifu Li
15 December 2019
Responses to Reviewer 2
1 Abstract (Line 22)
The unit of cm・a-1 is not generally used. Explanation of the abbreviation of “a” should be added.
Responses: Thanks very much for your helpful comment. We feel sorry for this unclear description. The unit of cm・a-1 means the annual average deposition rate. To make it more readable, we have changed “a” to “y” accordingly. Besides, we have changed all the units in the revised manuscript.
Please see Line 22, 149, 166, 169, 242-250, 260-261,265-269, 292, 301, 314, Figure 4 and Table 1. Meanwhile, the unit formats, and data identifiers in the Figure 2 and Figure 3 have been checked and modified uniformly.
2 Keywords (Line 30)
The word “cycle” is fairly abstract. The keyword should be more objective.
Responses: Thank you for your helpful comment. We revised it accordingly. The revised keyword is “Periodic response”.
Please see Line 30.
3 Materials and Methods
As the authors mentioned in line 96-97, sedimentation process in estuary is affected by human impact of upstream watershed and ocean process (wave and tide). Why authors deal with only upstream hydrology to evaluate the sedimentation process? The change of upstream area (land use or construction of cross drainage work) in the Liaohe river should be mentioned.
Responses: Thank you very much for your comments. We feel sorry for this unclear description.
(1) The main sediment source contribution area is the main stream basin of Liaohe River to the Liaohe estuary wetland tidal flat. Thus, we deal with only upstream hydrology to evaluate the sedimentation process. The data on land use changes in the main stream basin of the Liaohe River have been supplemented in materials and section.
Thanks for your proposal. The relationship between the tidal flat sediments of the Liaohe Estuary tidal flat and the land use change and human activities in the upstream of the Liaohe River basin will be a further research topics in the future. The relationship and mechanism between estuarine sediments and watershed changes will be explored in all aspects.
(2) We have added the data on the land use changes in the main stream basin of the Liaohe River in the section of Materials and Methods.
The revised descriptions are “According to the data of Resource and Environment Data Cloud Platform data of the Chinese Academy of Sciences (http://www.resdc.cn/data.aspx?DATAID=283), the main sediment source contribution area of the Liaohe estuary wetland tidal flat is the main stream basin of Liaohe River. During 1985 to 2017, the area of cultivated land, sandy land, construction land, and beaches and tidal flats increased by 4.36%, 84.0%, 85.0% and 29.5%, respectively; while the forest land, grassland, and water area decreased by 4.96%, 46.72%, and 9.52%, respectively[34].”
Please see Lines 102-108.
4 Materials and Methods (Line 108)
Is it right to recognize that the sampling points are not affected by any human influences? Condition of the sampling points should be described.
Responses: Thanks very much for your helpful comment. Based on the previous investigations and surveys of our research group, we can assure that the sampling points are not affected by any human influences. We have added the condition of the sampling points in the section of Materials and Methods.
The revised descriptions are “The sampling points located in the core area of the Liaohe Estuary National Nature Reserve, China, and human activity is not allowed in the whole region. The tidal flat was formed naturally, and there was no human disturbance in the past 50 years or more.”
Please see Lines 115-117.
5 Materials and Methods (Line 126)
The reference of Folk-Ward method should be added.
Responses: Thank you very much for your comments. We revised it accordingly.
Now the revised descriptions are “The sediment components and particle size parameters were statistically analysed and categorized using the Folk-Ward method [38].”
Please see Lines 134-135.
6 Materials and Methods (Line 162-175)
Calculation method of hydrological indicators should be accurately described. For example, the period of flood season or measuring method of the sediment discharge.
Responses: Thank you very much for pointing this out. We have added the calculation method of hydrological indicators in the section of Materials and Methods.
Now the revised descriptions are “The key hydrological indicators were measured and calculated as flows:
(1) The flow in the flood season: It refers to the river flow in the relatively concentrated rainfall season of each year, which was calculated according to the daily flow data of the river hydrological observation section. The flood season of Liaohe River Basin ranged from June to September in each year. The formula was as follow:
Eq (5)
Where is the flow in the flood season (m3·s-1), is daily average flow rate (m3·s-1), is the days of the flood season (D).
(2) The mean annual flow rate: It refers to the average value of river flow during each year, which was calculated based on the daily flow data of river hydrological observation sections. The formula was as follow:
Eq (6)
Where is the mean annual flow rate (m3·s-1), is daily average flow rate (m3·s-1), is the days of each year (D).
(3) The annual sediment discharge: It refers to the total sediment transported through the river hydrological observation section in one year, which was calculated from the daily sediment concentration data of river hydrological observation sections. The formula was as follow:
Eq (7)
Where is the annual sediment discharge (t·y-1), is daily average sediment concentration (kg·m -3), is daily runoff (m3), is the number of days in a year (D).
All the daily data was measured by automatic observation device, and then the mean daily data was automatically generated.”
Please see Lines 185-205.
7 Results (Line 246-252)
These sentences may move to discussion section.
Responses: Thanks very much for your helpful comment. We revised accordingly.
Please see Lines 369-373.
8 Results (Line 330)
The calculation method of correlation showed in Table 2 should be described in 2. Materials and Methods section.
Responses: Thanks very much for your helpful comment. We have added the the calculation method of correlation in the section of Materials and Methods.
Now the revised descriptions are “The correlation matrix of the Pearson correlation coefficient was used to analyze the correlations between hydrological indicators of Liaohe River and sediment particle size index. The analysis was completed in Matlab2010b software.”
Please see Lines 206-208.
9 Results (Figure 3)
As the results of dating analysis of 210Pb, the deposition rate was different among the sampling points especially after 1985 (such as deposition rate of C2 1985-1990 was steeper than C1). The differences of deposition characteristic among stations should be discussed.
Responses: Thank you very much for your comments. To better explain the finding in this study, we discussed them in the section of Discussion.
Now the revised descriptions are “In the present study, the deposition rate of the three sampling points was different, this may be due to that the dating results of 137Cs and 210Pb was not absolute date. Besides, the corresponding sedimentary time at the same mass depth was not the same, and resulted in different calculation results [43,44]. Furthermore, there was one year difference between the dating results of 137Cs and 210Pb, this implied that the dating results of the two methods were basically consistent and credible. Additionally, the average deposition rate of three sampling points ranged from 2.22 to 2.27 cm·y-1, with the maximum difference of 0.05 cm·y-1 and an average output value of 0.03 cm·y-1, this indicated that there was little differences in deposition rates, the values almost same in the area.
In addition, the deposition rate C2 was greater than C1 and C3, this may be caused by the extreme rainfall storm events in the upstream basin [45–47]. For example, there were extreme floods events of the Liaohe River in 1998 and 1993. The greater runoff capacity could transport a large amount of sediment to the relatively far distance of sampling point C2, and resulted in the little deposition in C1, while the larger distance of C3 leading the relatively little sediment. Therefore, there existed differences in the spatial distribution of deposition rate within the area.”
Please see Lines 362-369, 377-382.
Thanks again for your good suggestions. We deeply appreciate your hard work for our manuscript and your comments and suggestions are valuable to improve the quality of our manuscript.

Round 2
Reviewer 2 Report
The manuscript has been much improved and is in a nice condition now. I think this manuscript will be acceptable.